# Analysis of HPV-Positive and HPV-Negative Head and Neck Squamous Cell Carcinomas and Paired Normal Mucosae Reveals Cyclin D1 Deregulation and Compensatory Effect of Cyclin D2

**DOI:** 10.3390/cancers12040792

**Published:** 2020-03-26

**Authors:** Jiří Novotný, Veronika Bandúrová, Hynek Strnad, Martin Chovanec, Miluše Hradilová, Jana Šáchová, Martin Šteffl, Josipa Grušanović, Roman Kodet, Václav Pačes, Lukáš Lacina, Karel Smetana, Jan Plzák, Michal Kolář, Tomáš Vomastek

**Affiliations:** 1Laboratory of Genomics and Bioinformatics, Institute of Molecular Genetics of the Czech Academy of Sciences, 142 20 Prague, Czech Republic; jiri.novotny@img.cas.cz (J.N.);; 2Department of Informatics and Chemistry, Faculty of Chemical Technology, University of Chemistry and Technology, 160 00 Prague, Czech Republic; 3Institute of Anatomy, 1st Faculty of Medicine, Charles University, 128 00 Prague, Czech Republic; veronika.bandurova@lf1.cuni.cz (V.B.); karel.smetana@lf1.cuni.cz (K.S.J.); 4Department of Otorhinolaryngology, Head and Neck Surgery, 1st Faculty of Medicine, Charles University, Faculty Hospital Motol, 150 06 Prague, Czech Republic; 5Department of Otorhinolaryngology, 3rd Faculty of Medicine, Charles University, 100 00 Prague, Czech Republic; 6Institute of Microbiology of the Czech Academy of Sciences, 142 00 Prague, Czech Republic; 7Department of Pathology and Molecular Medicine, Charles University, 2nd Faculty of Medicine and Faculty Hospital Motol, 150 06 Prague, Czech Republic; roman.kodet@lfmotol.cuni.cz

**Keywords:** human papillomavirus, head and neck squamous cell carcinoma, cell cycle, D-type cyclins, *CCND1*, *CCND2*, *CCND3*, patient survival, paired tumor-normal samples, 11q13 amplification

## Abstract

Aberrant regulation of the cell cycle is a typical feature of all forms of cancer. In head and neck squamous cell carcinoma (HNSCC), it is often associated with the overexpression of cyclin D1 (*CCND1*). However, it remains unclear how *CCND1* expression changes between tumor and normal tissues and whether human papillomavirus (HPV) affects differential *CCND1* expression. Here, we evaluated the expression of D-type cyclins in a cohort of 94 HNSCC patients of which 82 were subjected to whole genome expression profiling of primary tumors and paired normal mucosa. Comparative analysis of paired samples showed that *CCND1* was upregulated in 18% of HNSCC tumors. Counterintuitively, *CCND1* was downregulated in 23% of carcinomas, more frequently in HPV-positive samples. There was no correlation between the change in D-type cyclin expression and patient survival. Intriguingly, among the tumors with downregulated *CCND1*, one-third showed an increase in cyclin D2 (*CCND2*) expression. On the other hand, one-third of tumors with upregulated *CCND1* showed a decrease in *CCND2*. Collectively, we have shown that *CCND1* was frequently downregulated in HNSCC tumors. Furthermore, regardless of the HPV status, our data suggested that a change in *CCND1* expression was alleviated by a compensatory change in *CCND2* expression.

## 1. Introduction

Head and neck squamous cell carcinomas (HNSCC) are malignant neoplasms that arise from the mucosal epithelial surface of the upper respiratory and digestive tracts. The worldwide annual incidence of head and neck cancers is more than 550,000 cases, thus making HNSCC the sixth-most common cancer in the world [1]. Unfortunately, the incidence of this tumor type continues to increase.

HNSCC is more prevalent in men than in women, and the majority of HNSCC cases occur in patients over the age of 60 [2]. The most affected subsites are the oral cavity, oropharynx, hypopharynx, nasopharynx and larynx. Early-stage, locally contained disease responds favorably to treatment, presenting very good cure rates of 70–90%. However, advanced HNSCC exhibit aggressive loco-regional invasions, frequent second primary tumors and lymph node metastases, while distant metastases are relatively rare. Despite advances in the treatment of HNSCC using molecularly targeted therapies, the five-year survival rate of loco-regional or recurrent/metastatic diseases has remained at 50–60% [3,4].

The main risk factors for the development of head and neck cancers include tobacco exposure and alcohol consumption, which are associated with more than 70% of all HNSCC cases [5,6], and the infection of high-risk oncogenic types of human papillomavirus (HPV) [7]. Among HPV high-risk types, HPV16 and HPV18 are the most common, accounting for more than 85% of all HPV-positive (HPV(+)) tumors [8,9]. HPV status, combined with the traditional tumor-node-metastasis (TNM) staging system, is considered to be a particularly significant biological prognostic marker and distinguishes two etiologically different subtypes of HNSCC [10]. Individuals with HPV(+) HNSCCs have a considerably better prognosis compared to those who possess HPV-negative (HPV(−)) cancers [11].

In HNSCC, the alterations which lead to functional loss of the tumor suppressor function are much more frequent than oncogene-activating mutations. These changes in both HPV subtypes almost invariably deregulate cell cycle entry and progression and, thus, are detrimental to cell proliferation. In HPV(−) tumors, p53 tumor suppressor is inactivated by genetic mutations, and retinoblastoma protein (pRb) is inactivated by cyclin D1 (*CCND1*) and the CDK4/6 complex. On the other hand, in HPV(+) tumors, viral proteins E6 and E7 inactivate p53 and pRb, thus enabling these tumors to evade cell cycle checkpoints [8,12]. Functional loss of p53, either by mutation or proteasomal degradation, occurs early in HNSCC development in approximately 75% of cases [13,14]. Likewise, another tumor suppressor gene, *CDKN2A*, that encodes cyclin-dependent kinase inhibitor p16^INK4a^, is frequently inactivated by a copy number loss in HPV(−) HNSCC [15,16]. In contrast, p16^INK4a^ overexpression is typical for HPV(+) HNSCC [17] and is used as a surrogate marker of HPV infection [13,16].

A mutation detected in negative regulators of the cell cycle is often paralleled by the increased expression of cyclin D1, a potent oncogene and positive regulator of the cell cycle that is encoded by the *CCND1* gene [15]. *CCND1* is a member of the D-type cyclin family that includes cyclin D2 (*CCND2*) and cyclin D3 (*CCND3*). Cyclins D1, D2 and D3 show roughly 50% of sequence identity at the amino acid level, and all have been shown to activate CDK4/6 and promote G1/S transition [18]. In nontransformed cells, the expression of D-type cyclins is controlled by mitogenic, adhesion and differentiation signals and, thus, integrates extracellular signals with the cell cycle. Overexpression of the D-type cyclins bypasses the requirement for mitogenic stimuli promoting unchecked proliferation and is believed to be an early cause of tumor formation [18,19,20]. Cyclin D1 protein overexpression has been reported in up to 70% of HNSCC cases [21,22], and gene amplification represents the most prominent mechanism of increased *CCND1* expression [15]. In addition, amplification of *CCND2* in HNSCC has also been reported [23]. Despite the frequent *CCND1* upregulation and its positive role in cell proliferation, the value of *CCND1* as an independent prognostic marker in HNSCC is inconsistent. Several studies have suggested that elevated *CCND1* expression correlates with poor prognosis of some types of HNSCC [21,24,25]; however, other studies have not found a significant correlation [26,27,28,29].

In the present study, we examined the expression of D-type cyclins in a new cohort of 94 HNSCC patients with known clinical data. For 82 of these patients, the expression profiles of both the tumor and matching normal mucosa were obtained using DNA microarrays. Analysis of expression changes between tumor and normal tissues revealed that a considerable fraction of tumors downregulates *CCND1* expression. However, the reduction of cyclin D1 expression did not correlate with a favorable clinical outcome. We also found that tumors with downregulated cyclin D1 expression frequently upregulate the expression of cyclin D2. Taken together, these results suggest a compensatory mechanism where the upregulation of cyclin D2 may be a direct consequence of the loss of related cyclin D1 function.

## 2. Results

### 2.1. Patient Characteristics

Normal (n = 86) and tumor (n = 90) samples were obtained from a total of 94 HNSCC patients from which clinical and pathological data were collected (Table 1 and Appendix A). Overall, patients were users of tobacco (81%) and alcohol (59%). HPV was detected in 26 cases (28%). In contrast to previous reports [30,31,32], there was almost no difference in the average age of HNSCC diagnosis between HPV(−) and HPV(+) subgroups. Primary treatment consisted of surgery alone in 15 (16%) cases, surgery combined with adjuvant radiotherapy in 73 (78%) cases and surgery combined with adjuvant chemoradiotherapy in 6 (6%) cases. Median follow-up length was 110 months for HPV(+) patients and 29 months for HPV(−) patients. HPV infection was associated with significantly longer overall survival (OS) (Figure 1A, left panel). Five-year OS was 77% (CI 62–95%) and 32% (CI 22–48%) in HPV(+) and HPV(−) groups, respectively. Hazard ratio (HR) of HPV(−) vs HPV(+) 2.13–9.15, *p* < 0.001). Disease-free survival (DFS) was also shorter for HPV(−) patients (five-year survival 61%; CI 46–81%) than for HPV(+) patients (five-year survival 85%; CI 72–100%) (Appendix A). However, the difference between HPV(−) and HPV(+) patients did not reach statistical significance (HR = 2.89, CI 0.95-8.77, *p* = 0.06).

We next analyzed whether HPV infection results in better prognosis of patients with the same tumor location. We analyzed tumors from the tonsils and the base of the tongue, as these subsites were significantly represented in respect of both the number and the proportion of HPV(+) tumors. Similar to the entire cohort, HPV infection of tonsil tumors was associated with significantly longer overall survival (Figure 1A, right panel). Five-year OS was 69% (CI 49–96%) and 43% (CI 18–100%) in the HPV(+) and HPV(−) groups, respectively. The hazard ratio of HPV(+) tonsil tumors was 0.3 (CI 0.09–0.99, *p* < 0.05). A similar trend was observed in patients with tumors originating in the base of the tongue, although in this case, statistical significance was not reached (Appendix A). Five-year OS was 71% (CI 24–100%) and 29% (CI 9–92%) in the HPV(+) and HPV(−) groups, respectively. The hazard ratio of the HPV(+) base of the tongue tumors was 0.27 (CI 0.07–1.11, *p* = 0.07).

Whole genome expression profiles were obtained for a cohort of 94 patients (90 tumor samples, 86 normal tissue samples). From this cohort, a group of 82 patients had expression profiles from both tumor and matched normal tissues. Of these 82 patients, 25 and 57 individuals had HPV(+) and HPV(−) tumors, respectively.

### 2.2. Deregulation of Cyclin D1 Expression in HPV(−) and in HPV(+) HNSCC Tumors Does Not Correlate with Patient Outcomes

Analysis of expression profiles from the cohort of 94 HNSCC cases revealed that the mean expression of *CCND1* does not differ between the normal mucosae and carcinomas. However, *CCND1* expression in tumors showed a visible spread toward both increased and decreased values (Figure 1B). Decreased *CCND1* expression was significant mainly in the HPV(+) group. We thus determined the changes in expression of *CCND1* between 82 paired tumor and normal tissues patient by patient. We considered *CCND1* to be deregulated in the tumor if the fold-change of expression between tumor and normal tissues was above two (*CCND1* upregulation) or less than 0.5 (*CCND1* downregulation), respectively. This categorization resulted in the formation of three different clusters: 47 cases (57.3%) in which there was no change in *CCND1* expression and 15 cases (18.3%) where *CCND1* was upregulated as expected in proliferating tumor tissue. However, a substantial number of tumors downregulated *CCND1* (20 cases, 24.4%) (Figure 1C). Further stratification according to HPV status revealed that *CCND1* downregulation was typical for HPV(+) tumors (13 cases, 52.0%), while almost all of the remaining 11 (44.0%) HPV(+) tumors did not deregulate *CCND1* expression. Only in one HPV(+) case (4.0%), *CCND1* was found to be upregulated (Figure 1C). On the other hand, HPV(−) tumors were spread across all three categories: tumors with *CCND1* expression upregulated (14 cases, 24.6%), unchanged (36 cases, 63.2%) or downregulated (7 cases, 12.3%). We examined whether these changes in *CCND1* expression correlated with patient outcomes and clustered 76 patients with known follow-up according to their *CCND1* regulation. Survival analysis revealed that neither OS (Figure 1D, Table 2 and Appendix A) nor DFS (Appendix A and Appendix A) were affected by *CCND1* upregulation in the HPV(−) group. *CCND1* downregulation was associated with a shorter five-year OS (Figure 1D and Table 2); however, the clinical data for this group was available for four patients only, showing no statistically significant difference (Table 2 and Appendix A). In the HPV(+) group, *CCND1* downregulation did not affect OS or DFS significantly (Figure 1E, Appendix A and Table 2). This data suggested that there is no significant association between worse clinical outcome and *CCND1* upregulation and, conversely, between better outcomes in patients with *CCND1* downregulation.

### 2.3. Cyclin D1 Upregulation Correlates with Bona Fide Amplification of Its Genomic Locus

The observed low number of tumors with upregulated *CCND1* expression was unexpected, as *CCND1* overexpression was generally reported in up to 70% of patients at the protein level [21,22]. The common mechanism underlying *CCND1* upregulation is its gene locus amplification, with frequency ranging from 17% to 50% of cases [33,34]. While we do not possess direct data on the amplification, we analyzed changes in the expression of genes surrounding the *CCND1* gene as its surrogate marker [15,22]. Locus amplification would lead to upregulation of other genes in the locus, together with the *CCND1* upregulation. Out of 20 genes present in the two-megabase window around the *CCND1* genomic locus, we could analyze changes in the expression of 12 genes, while the expression of the other six genes was not detected by the microarray technology (Figure 2A). In the group of 15 predominantly HPV(−) cases where *CCND1* expression was upregulated in the tumor tissues (FC > 2), we observed upregulation of the genes that were downstream of *CCND1* in 14 (93%) cases (Figure 2B,C). Notably, *ANO1* was upregulated both in the majority of HPV(−) tumors and in HPV(+) tumors, probably reflecting the fact that this protein is a potential driver oncogene [35]. With the exception of *TPCN2*, the genes upstream of *CCND1* were not upregulated along with *CCND1* in HPV(−) tumors. The expression of *MRGPRF* and *MYEOV* were rather downregulated (Figure 2B,C). The downregulation of *MYEOV* may be explained by frequent epigenomic silencing of the gene [36]. In summary, we observed a strong signal for amplification of the *CCND1* genomic loci in HPV(−) tumors. In the HPV(+) group, the co-amplification was observed only in one case, in agreement with previously published data that amplification of the *CCND1* locus is rare in HPV(+) tumors [22].

### 2.4. Deregulation of Cyclin D2 Does Not Correlate with Patient Outcomes

The expression levels of other D-type cyclins, *CCND2* and *CCND3*, were also examined. In contrast to *CCND1*, the median expression of *CCND2* was slightly elevated, yet it was statistically significant in both HPV(−) and HPV(+) tumors (Figure 3A). Comparative analysis of *CCND2* gene expression between paired tumor and normal samples revealed upregulation of *CCND2* expression in 26 cases (31.7%) (Figure 3B). In 49 cases (59.8%), there was no change in *CCND2* expression, and downregulation of *CCND2* was observed in seven cases (8.5%) only. Stratification according to HPV status revealed that both HPV(+) and HPV(−) tumors displayed significant upregulation of *CCND2* (Figure 3B), while downregulation of *CCND2* was observed only in a few HPV(−) tumors and a single HPV(+) tumor. Neither *CCND2* upregulation nor downregulation affected the OS (Figure 3C,D and Table 2) and DFS (Appendix A).

The analysis of *CCND3* mean expressions revealed elevated *CCND3* expression only in HPV(+) tumors (Figure 4A). Expression changes between paired tumor and normal samples also revealed the upregulation of *CCND3* expression in four cases (6.6%), predominantly in HPV(−) tumors (Figure 4B). Downregulation of *CCND3* was not observed. Due to the small number of patients with *CCND3* expression changes, statistical analysis did not yield any significant results (Table 2 and Appendix A).

### 2.5. Cyclin D2 is Often Upregulated in Tumors with Downregulated Cyclin D1 and Vice Versa

The finding that *CCND1* was downregulated in more than a fourth of the cases (Figure 1C) and that these patients did not display a better prognosis (Table 2) suggests the hypothesis that additional cyclins could compensate for the *CCND1* deficiency. We thus examined whether *CCND1* mRNA downregulation is paralleled by upregulation of *CCND2* and/or *CCND3*. Indeed, the comparison revealed that a substantial number of tumors with downregulated *CCND1* upregulated *CCND2* (7 out of 20; 35%) and *CCND3* in one individual case (Figure 5A). Intriguingly, we also found that tumors that upregulated *CCND1* often downregulated *CCND2* expression (5 out of 15; 33.3%) (Figure 5A). This observation suggests that expression of *CCND1* and *CCND2* is coupled.

### 2.6. Validation in an Independent Cohort

To validate our findings in an independent cohort, we analyzed the expression profiles from a cohort of 490 patients from The Cancer Genome Atlas (TCGA) obtained from cBioPortal (see Materials and Methods for details). In this cohort, there were 42 paired tumor and normal mucosa samples with follow-up clinical data; however, only three of these pairs contained HPV(+) tumors. Hence, we focused on HPV(−) pairs. Analysis of *CCND1*, *CCND2* and *CCND3* expression recapitulated the findings in our dataset. *CCND1* deregulation between tumor and normal tissues clustered patients in three groups (Figure 5B) with no correlation with OS and DFS (Appendix A). Substantial fraction of patients had *CCND2* in tumors upregulated (20 cases, 50.0%) or downregulated (12 cases, 30.0%) (Figure 5B). Importantly, we observed again that a significant fraction of the tumors with upregulated *CCND1* had downregulated *CCND2* (6 out of 9; 60.0%) or vice versa (7 out of 10, 70.0%) (Figure 5C).

### 2.7. Immunohistochemical Analysis of Tumor Samples

To evaluate the expression of cyclins at the protein level, we performed immunohistochemical analysis of 10 independent tumor samples (Figure 6). We observed cyclin D1 focal staining with variable intensity in the cytoplasm (10/10) and in the nuclei as well (9/10) (Figure 6A,B). The expression of cyclin D2 was weaker, and it was completely negative in three samples (Figure 6A); the nuclear signal of cyclin D2 was observed only in three cases (3/10) (Figure 6B). Frequently, we observed that the nuclear signal of cyclin D1 was accompanied by low or negative nuclear signals of cyclin D2 (Figure 6C,D). Vice versa*,* in the regions with high nuclear staining of cyclin D2, we observed that the level of cyclin D1 was low (Figure 6E,F).

## 3. Discussion

Amplification of the *CCND1* gene encoding cyclin D1 is one of the most frequent genomic alterations in human cancers [37]. Altered *CCND1* expression has been reported in many different cancers [18,19], including head and neck squamous cell carcinoma [13,38]. In this work, we have focused on the expression patterns of cyclin D1, as well as the other two members of the cyclin D family, in a new cohort of 94 HNSCC patients. This dataset contains expression data from 82 tumor samples and paired normal mucosa and, to our best knowledge, represents the largest collection of paired HNSCC samples. Analysis of expression changes between paired tumor and normal samples revealed that expression of the *CCND1* gene is upregulated mainly in HPV(−) patients, and, unexpectedly, it may also be downregulated, predominantly in HPV(+) patients. Nonetheless, there was no statistically significant correlation between *CCND1* deregulation and the disease outcome. Further analysis showed that *CCND1* upregulation was frequently accompanied by *CCND2* downregulation, and, conversely, its downregulation was found to be often accompanied by upregulation of *CCND2* or, in one case, of *CCND3*. Based on these data, we speculate that upregulation of one D-type cyclin could be compensated by downregulation of another one. Such compensatory effects may explain previously observed poor correlations between *CCND1* expression and patient prognosis [26,27,28].

The importance of D-type cyclins for tumorigenesis has been demonstrated in mouse models, where the presence of cyclin D proteins is required for both tumor initiation and maintenance, while they seem largely dispensable for normal development [39,40]. It is presumed that the increase in the cyclin D expression is an early oncogenic event that causes tumor formation and continuous uncontrolled proliferation of tumor cells [18,19,41]. Concordantly, with these studies, the analysis of almost 500 HNSCC samples showed amplification of chromosomal region 11q13 containing the *CCND1* gene in 31% of cases [15], although the *CCND1* amplification frequency has been reported to range from 17% to 50% [33,34]. The amplified 11q13 locus harbors several other genes that could promote tumorigenesis. Among them, *ANO1* coding for anoctamin-1 and *CTTN* coding for cortactin are frequently coamplified with *CCND1*. These genes were shown to promote HNSCC progression and to be associated with poor prognosis [22,29,35,42]. In agreement with that, we found that *LTO1, ANO1* and *CTTN* are coregulated in the HPV(−) tumors, and they could cooperate with *CCND1* to affect the clinical outcome.

In addition to gene amplification, other mechanisms likely contribute to cyclin D1 protein upregulation, as its overexpression estimated by immunohistochemistry has been documented in up to 70% of HNSCC tumors [21,22]. In our dataset, we observed *CCND1* mRNA upregulation in approximately 20% of cases, and, surprisingly, roughly 25% of tumors showed a decreased expression of the *CCND1* gene when compared to normal mucosa. Since the repression of cyclin D1 promotes cell cycle exit, cellular quiescence and, in some cases, cell differentiation [43] and, in HNSCC tumor models, reduces cell growth and survival [44], these findings indicated that a *CCND1* decrease may have a positive impact on patient outcome. However, our data did not show any statistically significant correlation between *CCND1* expression and disease outcomes.

Why patient outcome does not correlate with the changes in *CCND1* expression remains unknown. According to our data, we speculate that a functional significance of change in *CCND1* expression could be alleviated by the compensatory change in the expression of another D-type cyclin. Compensation among D-type cyclins has been observed in mice, where cyclins D1, D2 and D3 are expressed redundantly in most tissues. Genetic ablation of individual D-type cyclin does not generally affect normal development; however, the simultaneous ablation of cyclins resulted in more severe developmental defects and embryonal lethality. These studies suggested that D-cyclins can compensate for the loss of another family member [45]. A compensatory effect was notably evident in mice engineered to express only one D-type cyclin. In embryos, the sole expressed cyclin became upregulated in most tissues which were otherwise negative for this type of cyclin [46]. Cyclin compensation was also observed in adult mice in the proliferating uterine epithelium, where the cyclin D1 absence can be compensated by cyclin D2 [47], and in B-lymphocytes, where D2 absence is compensated by upregulation of cyclin D3 [48]. All these observations point to a mechanism where perturbation of one cyclin D member may induce the compensatory expression of other family members and correspond to our findings, where one-third of tumors with downregulated cyclin D1 mRNA upregulated the expression of cyclin D2. We also observed the opposite trend, where in one-third of tumors with cyclin D1 upregulation, cyclin D2 was downregulated, further supporting the hypothesis that genetic compensation in response to the perturbation of cyclin D function is a common phenomenon in HNSCC. We have validated our results in an independent TCGA dataset and also by immunohistochemical staining.

The best prognostic factor for HNSCC remains to be HPV infection, as HPV(+) patients display better clinical outcomes. The worse prognosis of HPV(−) patients may be the consequence of different mutational burdens in HPV(+) and HPV(−) HNSCC tumors [38]. It may also reflect the fact that the *CCND1* locus is rarely amplified in HPV(+) tumors [22,49], which is consistent with our findings that a positive correlation of expression of *CCND1* and the genes in the 11q13 locus is observed almost exclusively in HPV(−) tumors. The genes coamplified with *CCND1,* such as *LTO1/ORAOV1, ANO1* and *CTTN,* are associated with HNSCC metastasis and recurrence and may therefore account for the worse prognosis of HPV(−) patients [22,29,42,50,51]. Our observations that cyclin D1 expression is downregulated in half of HPV(+) tumors was surprising. Given that cyclin D1 is required for tumor formation induced by HPV proteins E6 and E7 [52], it indicated that cyclin D1 downregulation may also contribute to a better prognosis of HPV(+) patients. Indeed, previous studies have associated low cyclin D1 expression in HPV(+) tumors with improved survival rates when in combination with high p16^INK4a^, low pRb and low p53 [53,54]. However, we have not observed an improved outcome in HPV(+) patients who exhibit low expressions of cyclin D1 mRNA. In addition to the proposed compensatory effect of cyclin D2, it is also possible that the decrease in *CCND1* expression in HPV(+) tumors is a consequence of low pressure on cyclin D expression and lower CDK4/6 activity in general, as the G1 transcriptional repressor complex pRb/E2F is already disrupted by the viral protein E7 [14,38]. It is of note that, in HPV-associated cervical tumors, D-type cyclins and CDK4/6 activity inhibition by p16^INK4a^ promotes tumor cell survival [55]. It is possible to speculate that cyclin D1 downregulation contributes to the survival of HPV(+) HNSCC tumor cells, similarly to cervical cancer.

In conclusion, in our study with independent validation, we have analyzed the mRNA expression level changes between paired tumor and normal samples at the patient level. Availability of paired samples provided us with the opportunity to show a significant variation in D-type cyclin gene expression between individual patients and to further stratify the patients according to the changes in the expression of the D-type cyclins. Although our results do not support a role for a specific D-cyclin in patient prognosis, it is possible to speculate about a compensatory mechanism between D-type cyclin expressions. These results are in agreement with the current opinion that tumors can be addicted to cyclin D-dependent CDK4/6 kinase activity [56] and that CDK4/6–cyclin D complexes represent promising targets in cancer therapy [18,20,41]. In the clinical trials conducted in HNSCC patients with radioresistant recurrent tumors, a combination of cetuximab and palbociclib targeting the EGF receptor and CDK4/6, respectively, indeed show promising results [57,58]. Analysis of cyclin expression changes between patient tumors and normal tissues may lead to the employment of more effective personalized therapies that could enhance the efficiency of currently adopted cancer treatment regimens.

## 4. Materials and Methods

### 4.1. Sample Collection

Sample collection, detection of HPV and microarray analysis were performed as described in Valach et al. [59] and Szabo et al. [60]. In short, normal mucosa and tumor tissue specimens were collected from 94 patients suffering from HNSCC, after their informed consent in full agreement with the local ethical committee and the Declaration of Helsinki. Specimens were chemically protected from RNA degradation and stored at –85 °C. Detection of high-risk oncogenic HPV types (16, 19, 31, 33 and 45) in tumor samples was done by RT-qPCR of the *E6* and *E7* genes.

For immunohistochemical analysis, the tissues were fixed in 10% neutral buffered formalin and routinely processed for paraffin block preparation. Tissue sections (5-μm thick) were deparaffinized in xylene and ethanol baths with decreasing concentrations of ethanol (5 min each). Heat-induced epitope retrieval was performed at pH = 6.0 (citrate buffer). Sections were blocked in hydrogen peroxide blocking reagent and protein block (ab64218 and ab64226, respectively; both Abcam, Cambridge, UK). Primary antibodies (cyclin D1, clone DCS-6, mouse monoclonal; Dako, Glostrup, Denmark and cyclin D2 (D52F9) rabbit monoclonal #3741; Cell Signaling Technology, Danvers (MA), USA) were diluted 1:100, and sections were incubated at room temperature for 2 hours followed by incubation with HRP-tagged secondary antibody (Histofine simple stain MAX PO MULTI; Nichirei Biosciences, Tokyo, Japan) for 15 minutes. The reaction was developed for 10 minutes using Histofine simple stain AEC solution (Nichirei Biosciences, Tokyo, Japan) and counterstained with Gill´s hematoxylin (Sigma Aldrich, Prague, Czech Republic). For the negative control, the primary antibody was replaced by control nonimmune serum (X0910 and X0903; Dako, Glostrup, Denmark). Imaging was performed with a Leica DM2000 microscope using the LAS software package (Leica Microsystems GmbH, Wetzlar, Germany). The annotation of immunohistochemical results was performed using the method described in detail in Protein Atlas version 19.2 (https://www.proteinatlas.org/about/assays+annotation).

### 4.2. Transcription Profiling

Whole genome transcription profiling was performed on HumanWG-6 v3 Expression BeadChip microarrays (Illumina, San Diego, California, USA). Subsequent data analysis was done in R statistical environment [61]: raw data was processed using the limma package [62] of the Bioconductor Project [63]. Background corrected and quantile normalized data were corrected for batch effects using the ComBat function from the R package sva [64], and expression intensities of technical replicates were averaged at the probe level. Detected transcripts were annotated using the provided manifest file (HumanWG-6_V3_0_R2_11282955_A.bgx; Illumina, San Diego, CA, USA). Expression changes between the sample groups (tumor HPV(−), tumor HPV(+) and normal) were detected by the moderated *t*-test within the limma package.

### 4.3. Clinical Data

The following patient data was recorded: TNM stages, tumor grade and localization, presence of keratinization, extracapsular extensions, angiogenesis and perineural spread. Information regarding treatment was also recorded (surgery and adjuvant chemo-/radiotherapy), together with patient characteristics (smoking, alcohol usage, personal and family cancer anamnesis). Vital status was recorded by the clinician/investigator at time of last follow-up. For overall survival, vital status was alive or dead. For disease-free survival, four statuses were recorded: disease free, exitus of other reason, recidive and exitus of the disease. The latter two vital statuses were taken as events in the disease-free survival analysis. Association between the HPV(−) and HPV(+) groups and clinical variables was tested using either a Kruskal-Wallis or Fisher’s exact test.

### 4.4. Analysis of CCND1 Coamplification

To verify the co-amplification of the 11q13 region in HPV(−) tumors, we calculated log_2_ fold-changes of genes near *CCND1* (one megabase upstream and downstream of the *CCND1* start and end, respectively). The intensity of the probes targeting a particular gene was averaged, and genes with a mean probe log_2_-intensity lower than 5 were omitted. We used Gviz [65] and ComplexHeatmap [66] packages of the Bioconductor Project for visualization.

### 4.5. The Cancer Genome Atlas (TCGA) Validation Dataset

To provide an external validation, we used the head and neck squamous cell carcinoma TCGA PanCancer RNA-seq dataset of 490 HNSCC patients [15,67], provided through cBioPortal [68,69]. Read count matrices, clinical and follow-up data were downloaded and imported to the R statistical environment. DESeq2 [70] package was used for sequencing depth and variance-stabilizing normalization. DESeq2 was also used for statistical testing of the mean difference between sample groups (normal, tumor HPV(−) and tumor HPV(+)) using the Wald test.

### 4.6. Survival Analysis

We calculated fold-changes (FC) in expression of the D-type cyclins between tumor and normal mucosa samples for each patient and split the patients in three groups depending on FC: patients with cyclin upregulation (FC > 2), no deregulation (0.5 < FC < 2) and downregulation (FC < 0.5). These groups were used in survival analyses using the R packages survival [71] and survminer [72]. For each D-type cyclin, univariate Cox proportional hazard model, Kaplan-Meier estimator and its associated log-rank test were computed, using patients with no deregulation of cyclin as a reference.

### 4.7. Data Availability

The present dataset is available in MIAME-compliant form from the ArrayExpress database under accession E-MTAB-8588. The dataset used to validate our findings is available from The Cancer Genome Atlas via cBioPortal (https://www.cbioportal.org).

### 4.8. Ethics Approval and Consent to Participate

Ethical approval for patient recruitment, sample collection, clinical follow-up and data analysis based on the Declaration of Helsinki was granted by the Ethics Committee for Multi-Centric Clinical Trials of the University Hospital Motol and 2nd Faculty of Medicine, Charles University in Prague (Approval No EK-890/15).

## 5. Conclusions

In this study, we examined the expression of cyclin D1 and other D-type cyclins in a new cohort of HNSCC patients. The availability of both tumor sample and normal tissue sample from the same individual allowed us to analyze the relative changes in D-type cyclin expression rather than their absolute expression levels. The comparison of the D-type cyclin expression in tumor and healthy tissue from the same patients revealed an unexpected pattern of their expression. A subset of tumors increased the cyclin D1 expression that was specific, with one exception, to HPV-negative tumors. Unexpectedly, we also observed that the cyclin D1 expression was often downregulated in some tumors, with a higher frequency observed in HPV-positive patients. We have not found any direct effect of the changes in D-type cyclin expression on patient prognosis, even after stratification of the patients according to their HPV status. This observation could be a consequence of cyclin D2 upregulation, which frequently compensates cyclin D1 downregulation and vice versa. The compensatory expression among D-type cyclins was also observed in an independent dataset obtained from The Cancer Genome Atlas, further validating our data. We thus propose that absent correlation between the cyclin D1 expression and patient survival in both subtypes of HNSCCs may be a consequence of a compensatory mechanism where the effect of change in the cyclin D1 expression could be alleviated by the reciprocal change in the expression of cyclin D2. These findings highlight the importance of analyses of matched tissues from the same individual, as they can reveal molecular changes associated with the cancer development.

## Figures and Tables

**Figure 1 cancers-12-00792-f001:**
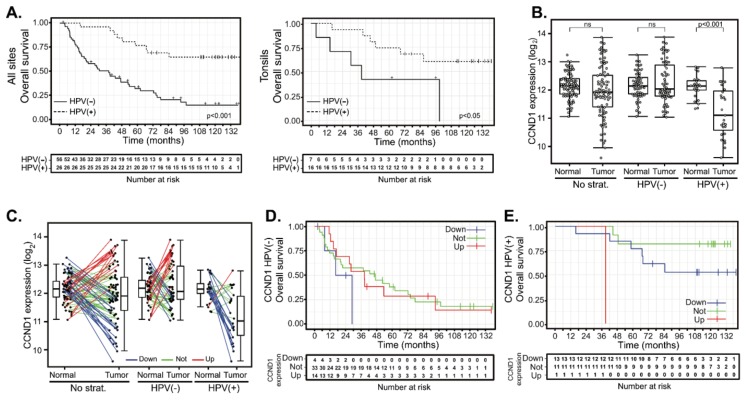
*CCND1* gene expression changes between tumor and normal tissues and patient survival in human papillomavirus (HPV) (+) and HPV(−) groups. (**A**) Kaplan-Meier plots of overall survival (OS) for HPV(−) and HPV(+) groups: entire cohort (left panel, n = 82) and tonsillar carcinoma (right panel, n = 23). (**B**) *CCND1* gene expression in a cohort of 94 patients (left panel) and stratified according to the HPV status (middle and right panel). The boxplot displays interquartile range with the whiskers indicating the greatest and smallest observations, excluding outliers. (**C**) *CCND1* deregulation between paired tumor and normal tissues. Paired samples are connected by the lines color-coded according to the change of *CCND1* expression (red—*CCND1* upregulation, fold-changes (FC) > 2; green—no deregulation, 0.5 < FC < 2 and blue—*CCND1* downregulation, FC < 0.5). (**D**) Kaplan–Meier plots for OS stratified according to *CCND1* gene deregulation in the HPV(−) group. (**E**) Kaplan–Meier plot for OS by *CCND1* deregulation in the HPV(+) group. Up—group with *CCND1* upregulated, not—group without *CCND1* deregulation and down—group with *CCND1* downregulated. Tick marks and crosses in (**A**,**D**,**E**) indicate right censoring.

**Figure 2 cancers-12-00792-f002:**
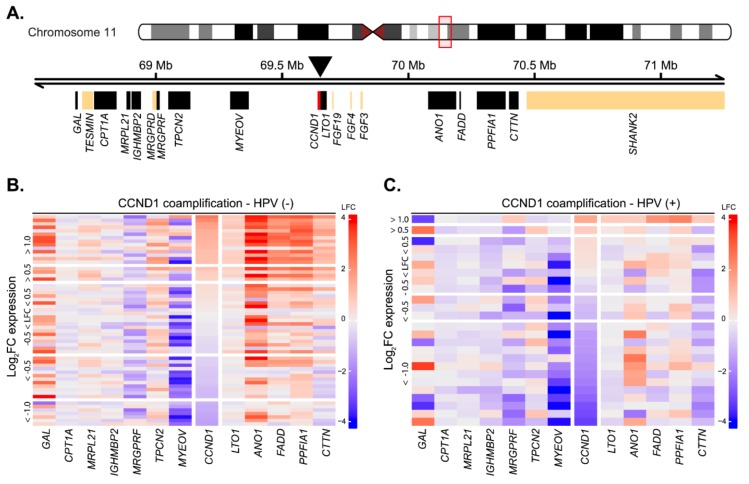
*CCND1* gene locus amplification as estimated from expression changes between tumor and normal tissues in HPV(+) and HPV(−) patient groups. (**A**) Schematics of the *CCND1* gene locus spanning ~ 2 Mb indicating position of the surrounding genes. Position of the *CCND1* gene (red) is indicated by the arrowhead. Black rectangles indicate expressed genes, and beige rectangles indicate genes expressed below the detection threshold. Position on chromosome 11 is given in Mb, million base pairs. (**B**) Heatmap of gene expression changes between tumor and normal tissues in the HPV(−) group for expressed genes from the *CCND1* locus. LFC—log_2_ fold-change in expression. (**C**) Heatmap of gene expression changes between tumor and normal tissues in the HPV(+) group for expressed genes from the *CCND1* locus.

**Figure 3 cancers-12-00792-f003:**
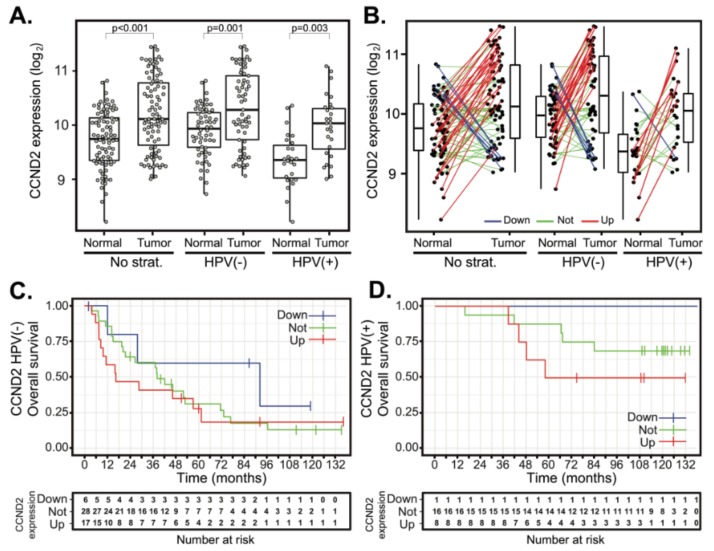
*CCND2* gene expression changes between tumor and normal tissues and patient survival in HPV(+) and HPV(−) groups. (**A**) *CCND2* gene expression in a cohort of 94 patients (left panel) and stratified according to the HPV status (middle and right panels). The boxplot displays interquartile range with the whiskers indicating the greatest and smallest observations, excluding outliers. (**B**) *CCND2* deregulation between paired tumor and normal tissues. Paired samples are connected by the lines color-coded according to the change of *CCND2* expression (red—*CCND2* upregulation, FC > 2; green—no deregulation, 0.5 < FC < 2 and blue—*CCND2* downregulation, FC < 0.5). (**C**,**D**) Kaplan–Meier plots for OS stratified according to *CCND2* gene deregulation in HPV(−) and HPV(+) groups. Up—group with *CCND2* upregulated, not—group without *CCND2* deregulation and down—*CCND2* group with *CCND2* downregulated. Tick marks in (**C**,**D**) indicate right censoring.

**Figure 4 cancers-12-00792-f004:**
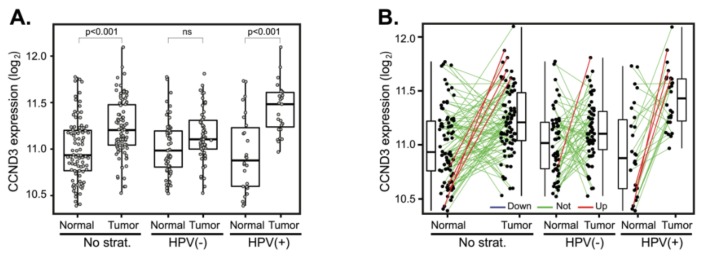
*CCND3* gene expression changes between tumor and normal tissues. (**A**) *CCND3* gene expression in a cohort of 94 patients (left panel) and stratified according to the HPV status (middle and right panels). The boxplot displays interquartile range with the whiskers indicating the greatest and smallest observations, excluding outliers. (**B**) *CCND3* deregulation between paired tumor and normal tissues. Paired samples are connected by the lines color-coded according to the change of *CCND3* expression (red—*CCND3* upregulation, FC > 2; green—no deregulation, 0.5 < FC < 2 and blue—*CCND3* downregulation, FC < 0.5).

**Figure 5 cancers-12-00792-f005:**
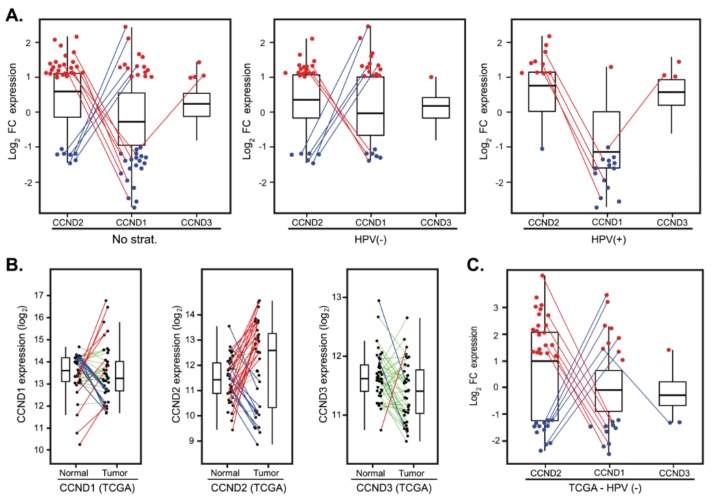
Compensatory expression of *CCND2* in tumors with deregulated *CCND1*. (**A**) D-type cyclins expression changes presented as log_2_ fold-change between tumor and normal tissues in a cohort of 82 patients (left panel) and stratified according to the HPV status (middle and right panels). For clarity, only tumors with FC > 2 (red dots) and FC < 0.5 (blue dots) are shown. Lines connect the same patients. (**B**) D-type cyclins deregulation between paired tumor and normal tissues in The Cancer Genome Atlas (TCGA) cohort of 39 HPV(−) patients. Paired samples are connected by the lines color-coded according to the change in D-type cyclin expression (red—upregulation, FC > 2; green—no deregulation, 0.5 < FC < 2 and blue—downregulation, FC < 0.5). (**C**) D-type cyclins expression changes presented as log_2_ fold-change between paired tumor and normal tissues in the TCGA cohort of 39 HPV(−) patients. For clarity, only tumors with FC > 2 (red dots) and FC < 0.5 (blue dots) are shown. Lines connect the same patients.

**Figure 6 cancers-12-00792-f006:**
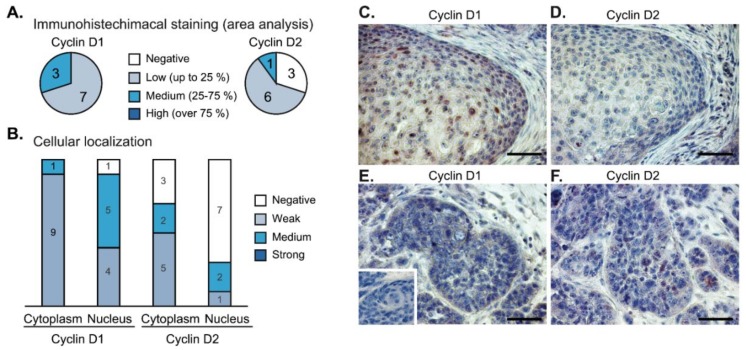
Immunohistochemistry of cyclin D1 and cyclin D2 expression in head and neck squamous cell carcinomas (n = 10). (**A**) Low-to-medium intensity staining of cyclin D1 was detected in all tumor samples. The staining of cyclin D2 was negative in three cases. (**B**) Nuclear and cytoplasmic distribution of cyclin D1 and cyclin D2 staining in the tumor samples. (**C**,**D**) In cases with high nuclear intensity of cyclin D1 (panel C), the nuclear staining of cyclin D2 was weak or negative (panel D). (**E**,**F**) Vice versa, higher intensity of cyclin D2 (panel F) was observed in tumor buds with a lack of nuclear cyclin D1 (panel E). Inset in panel E shows a negative control. The bar is 0.1 mm.

**Table 1 cancers-12-00792-t001:** Summary of clinical data for all patients in the present cohort (n = 94). For four patients, only samples from normal mucosa were available, and thus, the human papillomavirus (HPV) status was not determined. Statistical significance gauges the association between a variable and the HPV subtypes: (n.s.) non-significant, (*) *p* < 0.05 and (***) *p* < 0.001. Percentages are column-wise by default, row-wise if “r” is behind the value. (NA) not available.

Variable (Stat. Signif.)	All Patients N (%)	HPV(−) N (%)	HPV(+) N (%)
***No. patients***	94	-	-
***Sample groups***			
No. samples per normal mucosa	86	-	-
No. samples per tumor	90	64 (71r)	26 (29r)
***Patients***			
No. patients with paired samples	82	57 (70r)	25 (30r)
No. patients with paired samples and known follow-up	76	51 (68r)	25 (32r)
***Age at surgery (n.s.)***			
Median (range)	60 (26–94)	59 (26–94)	62 (41–72)
No. age < 40	1	1	0
***Gender (n.s.)***			
Female	10 (11)	6 (9)	4 (15)
Male	84 (89)	58 (91)	22 (85)
***Smoking (*)***			
No	17 (18)	7 (11)	9 (35)
Yes	76 (81)	56 (88)	17 (65)
NA	1 (1)	1 (2)	0
***Alcohol usage (n.s.)***			
No	38 (40)	22 (34)	14 (54)
yes	55 (59)	41 (64)	12 (46)
NA	1 (1)	1 (2)	0
***Tumour site (***)***			
base of the tongue	16 (17)	9 (14)	7 (27)
hypopharynx	4 (4)	4 (6)	0
larynx	21 (22)	20 (31)	0
oral cavity	21 (22)	18 (28)	1 (4)
oropharynx part	9 (10)	6 (9)	2 (8)
tonsils	23 (24)	7 (11)	16 (62)
***Stage (n.s.)***			
I	8 (9)	6 (9)	2 (8)
II	8 (9)	5 (8)	3 (12)
III	21 (22)	17 (27)	4 (15)
IV	56 (60)	35 (55)	17 (65)
NA	1 (1)	1 (2)	0
***Grade (*)***			
G1	19 (20)	18 (28)	1 (4)
G2	46 (49)	26 (41)	18 (69)
G3	28 (30)	19 (30)	7 (27)
G4	1 (1)	1 (2)	0
***Months of follow-up (median) (range) (n.s.)***	47 (0–139)	29 (0–138)	110 (16–139)

**Table 2 cancers-12-00792-t002:** Survival analysis, overall survival (OS). Patients with paired samples and known survival (n = 76, see Table 1) were clustered according to the deregulation of D-type cyclin expression in matching samples. Thresholds for deregulation were the following: not (0.5 < fold-changes (FC) < 2), up (FC > 2) and down (FC < 0.5). HR—hazard ratio and NA—not available.

	Five-Year Survival (%)		Univariate HR (95% CI), *P*	
	HPV(−)	HPV(+)	HPV(−)	HPV(+)
***CCND1***				
not	38	82	1	1
up	29	0	1.04 (0.5–2.17), 0.92	46.95 (2.26–973.39), 0.01
down	0	77	2.6 (0.86–7.85), 0.09	2.88 (0.58–14.3), 0.2
***CCND2***				
not	30	81	1	1
up	28	50	1.16 (0.59–2.29), 0.67	2.04 (0.54–7.69), 0.29
down	60	100	0.51 (0.15–1.72), 0.28	NA
***CCND3***				
not	32	77	1	1
up	100	67	NA	0.9 (0.11–7.22), 0.92
down	0	0	NA	NA

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
