# Peer review of "Analysis of HPV-Positive and HPV-Negative Head and Neck Squamous Cell Carcinomas and Paired Normal Mucosae Reveals Cyclin D1 Deregulation and Compensatory Effect of Cyclin D2"

_cancers, 2020, doi:10.3390/cancers12040792_

Round 1

Reviewer 1 Report

The authors demonstrated a microarray analysis using 94 head and neck squamous cell carcinoma (HNSCC) samples derived from multi-subsites to show the status of cyclin Ds expression in the context of human papillomavirus (HPV) positivity. This manuscript shows up and down-regulations of cyclin D1 (CCND1) in HNSCC, however it is not statistically correlated with patient outcome. Finally, they found a compensatory expression pattern between CCND1 and CCND2 in some HNSCC tumors regardless of the HPV status and confirmed a similar observation in public database (TCGA). Although the results are derived from microarray analysis based on mRNA expression, this manuscript will be further strengthened after reconsideration of the following points.

 Major:

Further analysis of CCND1, CCND2 and CCND3 expression statuses should be conducted in stratified primary tumor subsites to see the correlation of patient’s OS, and the compensatory pattern of CCND1 and CCND2 in each subsite. The compensatory expression pattern of CCND1 and CCND2 should be validated by protein levels, such as western blot and/or immunohistochemistry in tumor samples since all commercial antibodies are available.

Minor:

Percentage calculation will need to be corrected, especially at line 133. It will be useful to add primary tumor sites information in Table 1. Gene name in Figure 3 legend will need to be corrected.

Reviewer 2 Report

The study exanimes cyclin D1 and D2 expression in HNSCCs. Cyclin D1 was down-regulated and Cyclin D2 was upregulated as a compensatory mechanisms. The results are as they are and are in a way not unexpected and not so clear cut as one would have wished for. But that is how it is and not unusual for this kind of studies.

The methodology is sound and well described.   

Author Response

Reviewer 2

We would like to thank you and other reviewers for the comments and helpful analysis of our manuscript, which we believe has helped to strengthen our paper.

As described below, in the revised manuscript we incorporated results of additional experiments and analyses (Figure 2 and Figure 6) and rewrote the text accordingly. We also rewrote some parts of the manuscript to make the text clearer and to avoid the data over-interpretation and repetition. As a consequence, the figure and reference numbering has changed. We also revised the caption of Table 2 to better describe its content. All changes are highlighted in the revised manuscript. The grammar has been checked once again by a native speaker.

Below we provide a point-by-point response to your comments. We also include supplementary tables (Tables R1 – R3) that are intended for reviewing purposes only. This data has not been incorporated into the revised manuscript as we think it would not bring any substantially new information.

Response: Reviewer 2 did not raise any critical points. We thank her/him for the positive review.

Reviewer 3 Report

It is well known that higher cyclin D1 expression is associated with higher tumor stage and metastasis of HNSCC. In addition, compensation among D-type cyclins seems not a novel data. Although cases of low expression of CCND1 appear more frequently in HPV-positive tissues, they are not statistically significant with OS.
In addition, the title ‘HPV-dependent cyclin D1 deregulation’ was not confirmed. Above all, the number of samples in each group is not clearly defined. Especially, in Table 1, the differences between normal and tumor groups should be analyzed statistically.

Author Response

Reviewer 3

We would like to thank you and other reviewers for the comments and helpful analysis of our manuscript, which we believe has helped to strengthen our paper.

As described below, in the revised manuscript we incorporated results of additional experiments and analyses (Figure 2 and Figure 6) and rewrote the text accordingly. We also rewrote some parts of the manuscript to make the text clearer and to avoid the data over-interpretation and repetition. As a consequence, the figure and reference numbering has changed. We also revised the caption of Table 2 to better describe its content. All changes are highlighted in the revised manuscript. The grammar has been checked once again by a native speaker.

Below we provide a point-by-point response to your comments. We also include supplementary tables (Tables R1 – R3) that are intended for reviewing purposes only. This data has not been incorporated into the revised manuscript as we think it would not bring any substantially new information.

Comments and Suggestions for Authors

Comment: It is well known that higher cyclin D1 expression is associated with higher tumor stage and metastasis of HNSCC. In addition, compensation among D-type cyclins seems not a novel data.

Response: We respectfully disagree with the reviewer that compensation among D-type cyclins is not a novel observation. Although such compensatory mechanism has been explored in mouse models, to our best knowledge it has not been extensively characterized in clinical settings.

Comment: In addition, the title ‘HPV-dependent cyclin D1 deregulation’ was not confirmed.

Changes appear on pages and lines: pg. 1, 2

Response: We agree with the reviewer that the title was misleading and we changed its wording appropriately. The new title reads “Analysis of HPV-positive and HPV-negative head and neck squamous cell carcinomas and paired normal mucosae reveals cyclin D1 deregulation and compensatory effect of cyclin D2.”

Comment: Above all, the number of samples in each group is not clearly defined. Especially, in Table 1, the differences between normal and tumor groups should be analyzed statistically.

Changes: pg. 4, 138-140; pg. 13, 439-440

Response: For the vast majority of analyses we used paired tumor-normal samples, and thus there were no differences in clinical covariates. However, we performed appropriate statistical tests for the correlation of clinical variables with HPV subtypes and indicated the results in modified Table 1. The statistical approaches are described in the Methods section.

Reviewer 4 Report

This work analyzes the expression at mRNA level of the genes CCND1, CCND2 and CCND3 in a series of 94 head and neck carcinomas with known HPV status. The main findings are the discovery that a proportion of cases present decreased expression of CCND1, which seems to be counteracted by increased expression of the other cyclins (at least in a part of those cases). The findings are confirmed in an analysis of data obtained from the TCGA database. But these changes in the expression of the cyclin-D coding genes have no prognostic significance.

On the other hand, the work confirms the better prognosis of patients with HPV+ tumors. And they also find that HPV+ tumors less frequently present over-expression of CCND1 (only one case presented it). This is consistent with the absence of amplification of the CCND1 gene and overexpression of the cyclin D1 protein observed by other authors in HPV+ tumors (DOI: 10.3390/jcm7120501; work that should have been cited in the article).

Since one of the main mechanisms of cyclin D1 over-expression is amplification of the CCND1 gene (which has been consistently related to poor prognosis in HPV- carcinomas), in addition to analyzing the expression of this gene, they should have analyzed its amplification. And it would also have been interesting to analyze the expression of cyclins D1, D2 and D3 at the protein level, to see if there was a correspondence between mRNA levels and this one.

However, at the protein level, one would expect higher percentages of cyclin D1 overexpression, since (as indicated by the authors in the discussion), cyclin D1 overexpression is usually described in more than 50% of the cases. It is therefore surprising to find not only the downregulation of its expression, but also the small percentage of cases (18%) with over-expression found at the mRNA level.

The discussion speculates on the biological significance of the correlations between the different levels of expression of the cyclin-D coding genes. But since no experimental testing is done on the proposed hypotheses, they are still speculations.

In addition to these concerns about the work, the wording is somewhat repetitive: the first part of the discussion is a repetition of the results; the discussion ends with a brief conclusion, but then there is another section of conclusions in which the results are summarized again and conclusions are given. All this should be corrected in a way that is not so repetitive.

Round 2

Reviewer 1 Report

This revised version was significantly improved. 

Minor comment: Case numbers are not identical in multiple represented results in the manuscript, supplementary figure and tables for reviewer only. For examples, HPV (-) and HPV (+) in the table1 are 9 and 7, respectively. However, those numbers are not consistent in supplementary figure 1 as HPV(-)=8, in table R1 as HPV(-)=7, and HPV (-)=6 in table R3. Is there any specific reason(s) on this discrepancy?

Author Response

Reviewer 1

Comments and Suggestions for Authors

Minor comment: Case numbers are not identical in multiple represented results in the manuscript, supplementary figure and tables for reviewer only. For examples, HPV (-) and HPV (+) in the table 1 are 9 and 7, respectively. However, those numbers are not consistent in supplementary figure 1 as HPV(-)=8, in table R1 as HPV(-)=7, and HPV (-)=6 in table R3. Is there any specific reason(s) on this discrepancy?

Changes appear on pages and lines: pg. 3, 134; pg. 5, 168-169; pg. 6, 176; caption of Supplementary Figure S1

Response: Thank you for the comment. Actually, the numbers are correct: Table 1 shows data for all patients with the carcinoma of the base of the tongue (n = 16). The numbers are lower in Suppl. Fig. 1B as it shows only data for patients with known survival (n = 15, for one patient there was no follow-up). Similarly, in Table R1 we display data for patients with paired samples (n = 14), only, and in Table R3 we show data for patients with both known survival and paired samples (n = 13). We clarify the differences in the text to avoid misunderstanding.

Reviewer 3 Report

I think it is okay to accept it as it is.

Author Response

Reviewer 3 did not raise any further points on revised manuscript. We thank her/him for the review.

Reviewer 4 Report

The authors have adequately answered most of the questions raised. Although the requested study of the amplification of the CCND1 gene is done by an indirect estimation, it can be considered sufficient

Author Response

Reviewer 4 did not raise any further points on revised manuscript. We thank her/him for the review.